# An Analysis of Comparative Perspectives on Economic Empowerment among Employment-Seeking Survivors of Intimate Partner Violence (IPV) and Service Providers

**Sarah Tarshis** [1,*], **Heather Scott-Marshall** [2] **and Ramona Alaggia** [3]

1   School of Social Work, Carleton University, Ottawa, ON K1S 5B6, Canada
2   Social and Behavioural Health Sciences Division, Dalla Lana School of Public Health, University of Toronto, Toronto, ON M5S 1A1, Canada; heather.scott@utoronto.ca
3   Factor-Inwentash Faculty of Social Work, University of Toronto, Toronto, ON M5S 1A1, Canada; ramona.alaggia@utoronto.ca
*   Correspondence: sarah.tarshis@carleton.ca

**Abstract:** Purpose: The purpose of this study is to compare perspectives on economic empowerment in the context of employment seeking among intimate partner violence (IPV) survivors and service providers specializing in IPV-related trauma. This study addressed the following question is: How do employment-seeking IPV survivors and service providers conceptualize and understand empowerment? Insights into how each group conceives of economic empowerment and its attainment following IPV experiences can help to inform an effective service curriculum that can be used to facilitate optimal employment outcomes among survivors. Methods: A constructivist grounded theory method was used to develop a theoretical framework for conceptualizing how economic empowerment is understood by employment-seeking survivors of IPV, and IPV service providers. Twenty-six participants were recruited (survivors, *n* = 16; service providers, *n* = 10) in a large northeastern U.S. city. Interview questions focused on how IPV survivors and service providers identify and conceptualize economic empowerment, and how support services respond to survivors' needs around empowerment, particularly through help with employment seeking. Results: Data were coded and analyzed following data analysis stages: (a) initial coding; (b) constant comparison; and, (c) focused coding. Three main themes emerged from the narrative data: (1) structural characteristics shape individual experiences and perspectives of empowerment; (2) peer support as an integral component to empowerment; and (3) employment attainment as economic empowerment. Though perspectives on economic empowerment were often aligned, some key differences emerged. Whereas providers tended toward a more restricted, micro-level view of empowerment as primarily an attribute of the individual, survivors were inclined toward a structuralist perspective that considers how individuals' experiences of empowerment are shaped by broader, institutional-level factors. Conclusions: Findings from this study build on prior research on the experiences of IPV survivors. The focus on experiences of empowerment in the context of employment-seeking can inform work on building more effective support services for survivors that avoid reductionist approaches that could be perceived by survivors as "victim-blaming" by incorporating a sensitivity to empowerment as derived from structural factors that shape individual experience.

**Keywords:** intimate partner violence; economic abuse; economic empowerment; structural barriers

## 1. Introduction

Intimate partner violence (IPV) impacts one in every three women and continues to be a pervasive public health concern [1,2]. IPV is defined as the physical, sexual, coercive, economic and/or psychological harm committed against an individual by an intimate partner [2,3]. Economic abuse is an emerging field within IPV scholarship that has only recently been conceptualized as a distinct form of violence [4,5]. It describes abusive

actions that control a victim's ability to access and utilize financial resources and includes tactics such as limiting employment opportunities and access to finances, and interference in financial decision-making [6]. Economic abuse is common in abusive relationships yet remains understudied in IPV research [7,8]. Some scholars have described it as an "invisible" form of violence that impacts employment, education, finances, and general well-being [4,9].

Over the past decade, economic empowerment initiatives have been developed and implemented within IPV service organizations to help mitigate the effects of economic abuse, and to support the financial stability, security, and safety of survivors [10]. Economic empowerment has been defined as the process of obtaining the knowledge, skills, and beliefs needed to establish economic independence [6]. A central component is acquisition of key intra- and inter-personal skills namely, economic self-sufficiency (the ability to financially support oneself), self-efficacy (a set of beliefs linked to performance and activities), and financial literacy (knowledge and skills needed to make financial decisions) [11,12]. Research on economic empowerment and IPV has focused on forms of financial empowerment, such as financial literacy, asset building (e.g., micro-lending/-financing), and other skills generally aimed at enhancing financial security [8,13].

Studies indicate that support services focused on empowerment can be effective in increasing self-efficacy, self-esteem and confidence among IPV survivors [6,14–17]. To date, however, there remains a gap in knowledge regarding how survivors and providers understand empowerment, particularly within the context of employment-seeking by survivors. Findings from the study can inform evidence-based recommendations for economic empowerment-focused services for women who have experienced inter-personal violence and other forms of abuse. Using a qualitative approach, the present study seeks to better understand empowerment through an analysis of the comparative perspectives of IPV survivors who are actively seeking employment and the providers of IPV support services.

## 2. Research on Economic Empowerment and Career Counseling

Empowerment-focused interventions for IPV survivors are designed to address the adverse impacts of economic abuse and help individuals gain economic independence. These interventions focus on enhancing economic empowerment through financial literacy—i.e., provide knowledge and skills for making sound financial decisions [10,13]. The two main economic empowerment interventions currently in use are the Redevelopment Opportunities for Women's Economic Action Program (REAP) [10] and the Allstate Foundation's Moving Ahead Through Financial Management curriculum [13,18]. These interventions have been systematically studied for their contribution to individual outcomes in financial literacy and financial management. Findings from studies of these interventions show that IPV survivors who participated in structured financial literacy training reported increases in self-efficacy [10], greater knowledge of financial management [18], and increased levels of monetary savings [5]. As a consequence, an increasing number of interventions have taken a similar approach to economic empowerment by focusing on enhancing the financial literacy of individuals.

Other scholars have examined community career counseling and employment interventions within the field of vocational psychology [19,20]. The intervention known as ACCESS—Advancing Career Counseling and Employment Support of Survivors of Domestic Violence evaluates a short-term group career counseling workshop for survivors of IPV [14]. The model is designed with many original grassroots empowerment principles including raising awareness of power dynamics, gaining employment knowledge, empowerment [21]. In their study, the authors designed a career intervention program developed to support survivors with their careers and help economic stability [14]. Attention to survivors' employment, career development, and critical consciousness was found to promote well-being, career development, and individual and collective empowerment [14,22]. Moreover, the authors recommended that programs designed to enhance

survivors' economic stability should integrate a greater awareness of the impact of violence into intrapersonal and interpersonal knowledge, beliefs, and skills. These studies helped launch a fulsome program of research into the efficacy of economic empowerment and career focused interventions for IPV survivors.

## 3. The Impact of Neoliberal Policies on IPV Service Provision

Whereas empowerment frameworks favour services focused on collective empowerment, consciousness-raising, and social change, these tenets are increasingly supplanted by neoliberal policies that place a premium on maximizing efficiency in IPV support services. Moreover, these types of services view the individual as the focal point of treatment, largely divorced from broader life circumstances [23]. Scholars have highlighted the link between IPV services and neoliberalism particularly among survivors seeking support from state-funded social services (e.g., welfare, criminal justice system, IPV after-care and supports) [24,25]. Services that aim to empower survivors often co-opt the language of empowerment to perpetuate neoliberal discourse that foregrounds individual accountability [24,26,27]. This shift in ideology focuses on the individual, in this case the survivor, and their duty to make use of all available resources to improve their quality of life. As a result, experiences of empowerment become individualized and detached from community and social structure [23], alienating individuals from broader sources of empowerment.

The impact of neoliberal models on IPV service provision remains evident today and can be traced back particularly to the lasting effects of the "Personal Responsibility and Work Opportunity Reconciliation Act" (PRWORA) of 1996. PRWORA significantly impacted how survivors engage in welfare programs and seek employment and remains in effect through current welfare policies [25,28]. The federally funded program, Temporary Assistance for Needy Families (TANF) incentivized work by making receipt of welfare benefits contingent on individual recipients working part-time, often in low-paid jobs [6]. These requirements pose several problems. In welfare-to-work programs, survivors are required to work a set number of hours for a certain amount of income in order to receive welfare benefits. If survivors are unable to comply with the program policies, there are punitive measures in place (e.g., removal of benefit, sanctions) which puts them in a vulnerable position where they risk losing essential benefits such as food stamps, shelter, and childcare vouchers [29]. Research has shown that many of these programs do not help survivors find employment or achieve long-term economic stability [30,31]. Studies on welfare-to-work programs have found that many survivors occupy a permanent state of financial disadvantage as a result of being relegated to low-skill, low-paying jobs [30].

## 4. Theoretical Considerations

Existing literature on IPV largely ignores perspectives on the structural determinants of empowerment. Scholars have recommended that the research community rethink and retheorize empowerment to broaden its conceptualization beyond the individual to incorporate the structural elements of power and engagement in political action [14,24,32,33]. A social ecological perspective [34] has been widely used in IPV practice and research because it examines the interconnection between individual, relational, community, and structural factors [35].

An intersectional perspective applied to IPV service provision also brings into consideration the multiple aspects of survivors' lives, including their abuse experiences, social location, financial resources, and employment circumstances and how these shape economic well-being [36]. Intersectional feminist theory challenges the limitations of earlier feminist ideologies by further examining the unequal power relations in society and recognizing the intersection of social and political identities by age, gender, class, race, immigration status, and other identities [36,37]. Social factors such as age, race, class, sexual and gender identity, immigration status, among others, profoundly shape experiences of power, oppression, and empowerment practices. By making explicit the links between social position and inequality, this perspective strives for broader social change [37]. Intersectionality

also foregrounds the connection between knowledge, consciousness, and the politics of empowerment, while prioritizing the role of knowledge in empowering those who have experienced oppression [37].

*Major Objective of the Study*

Research is needed to determine how empowerment is conceptualized and understood among survivors of IPV and service providers; whether perspectives of IPV survivors and service providers are aligned or misaligned; and how to ensure that the needs of survivors who are seeking employment are met.

The primary objective of the present study is to contribute to the limited research in this field by: (1) exploring survivors' and service providers' perspectives on empowerment and employment-seeking; (2) comparing and contrasting their perspectives; and (3) highlighting evidence that points to effective approaches to empowerment and employment-related services.

This study addressed the following research questions:

Research question 1: How do employment-seeking IPV survivors and service providers perceive and define empowerment?

Research question 2: What do employment-seeking IPV survivors and service providers identify as experiences relevant to empowerment (or alternatively, disempowerment)?

Research question 3: How are the different identities of employment-seeking survivors linked to social location (as given by gender, race and sexual identity), and how do these structural factors impact experiences of empowerment?

Research question 4: What are the implications for empowerment practices for survivors of IPV seeking employment?

## 5. Methodology

This study utilizes both a social ecological and intersectional theoretical approach and employs a constructivist grounded theory methodology [38] to develop mid-level theory about how empowerment is conceptualized among employment-seeking IPV survivors and service providers. Specifically, this study aims to reconceptualize empowerment from the perspectives of both survivors and service providers using an integrated theoretical approach to generate a comprehensive understanding of empowerment and IPV.

## 6. Recruitment and Analysis

All research materials were approved by the University of Toronto Research Ethics Board. A multi-disciplinary IPV Community Advisory Board (CAB) was developed to consult with throughout the research process. The study was located in a large northeastern American city and participants were recruited through services in this jurisdiction.

### 6.1. Recruitment and Data Collection

Data collection consisted of two phases: Interviews with: (1) IPV survivors and (2) IPV service providers (see Tables A1 and A2). Both phases of data collection began with initial sampling [38], as an entry point to specific inclusion criteria. First, IPV survivors who were either employment-seeking or already employed were recruited. Theoretical sampling focused on ensuring that the sample represented survivors with diverse identities and backgrounds (e.g., LGBTQ, immigrants, justice system-involved) in order to develop a broader understanding of IPV and economic empowerment experiences. Study flyers were distributed to shelters, counseling, and social service settings providing IPV services throughout a large northeastern U.S. city. Telephone screening ensured that participants were 18 years of age or older, spoke English, and were either employed or actively seeking employment.

### 6.2. Data Analysis

The author conducted all interviews and data collection took place between March 2018 to March 2019. Written consent was obtained from all participants. Each participant filled out a demographic sheet. Interviews lasted 30 min to two hours, with an average of one hour and five minutes. No honoraria were offered to service providers though a $25 gift card and the costs of transportation were offered to those who were not service providers. All interviews were digitally recorded, transcribed verbatim, and uploaded into NVivo 12 qualitative software for analysis. Analysis proceeded iteratively, alternating between interviewing, transcribing, and analysis [38]. Line by line coding, constant comparison to highlight similarities and differences within and across cases, prolonged engagement with the data, memo-ing, and consultation with the IPV CAB were techniques used to ensure data trustworthiness [39]. Data source triangulation, comparing survivors' and service providers' data, also enhanced the trustworthiness of the study results. Focused coding to identify emerging theoretical concepts led to the identification of major themes [38]. In the final step, themes and subthemes from service providers and survivors were compared and contrasted to illustrate their shared and unique perspectives (see Table 1).

**Table 1.** Themes and subthemes of empowerment mapped across both groups of participants.

| Themes and Subthemes | Survivor Participants | Service Provider Participants |
|---|---|---|
| Structural characteristics shape individual experiences and perspectives of empowerment | | |
| Fluid definition of empowerment | x | x |
| Includes sense of safety, living violence free, confidence | x | |
| Sense of personal agency that is not oppressive | x | x |
| Goal-oriented | x | x |
| Conceptual terms (e.g., self-determination, self-sufficiency) | | x |
| Oppressive and problematic term | x | x |
| A process managed by service providers | x | x |
| Power imbalances | x | |
| Programs that disempower | x | x |
| Professional obligation to ensure survivors exercise agency | | x |
| Employment discrimination based on identity | x | x |
| Intersectional stigma and transphobia | x | x |
| Pervasive sexism in the workplace | x | x |
| Anti-Black racism and employment | x | x |
| Social change efforts | | x |
| Empowerment and peers support among employment-seeking survivors | | |
| Connecting with other survivors | x | |
| Sharing story of IPV | x | |
| Building community | x | x |
| Population-specific empowerment services | x | x |
| Survivor-led services | | x |
| Linking employment and economic empowerment | | |
| The benefits of engaging in economic empowerment programs (EEPs) | x | x |
| Employment is empowerment | x | |
| Not everyone gets a job after engaging in EEPs | x | |
| Agency as an important aspect in EEPs | x | x |
| Relational process of empowerment in EEPs | x | x |
| Peer support in EEPs | x | x |
| Social change in EEPs | | x |
| Importance of using different service strategies | | x |

### 6.3. Sample Description

This resulted in 16 survivors coming forward to participate in the study. The inclusion criteria included those who: (1) were over the age of 18; (2) no longer in the abusive

relationship; and (3) were either employment-seeking or already employed. Ten were employed and six were actively job-seeking and/or volunteering or interning. Survivors who were employed worked in various sectors (e.g., for profit, non-profit, and self-employed; food service, beauty, health) with an annual average income of $24,000. The mean age of participants was 36 years old, ranging from 24 to 50 years old. More than half (9) of the survivors had child-care responsibilities. The sample included cisgender women (13) and transgender women (3). Fifteen survivors identified as heterosexual and one identified as queer. Eleven women were born outside the U.S. and five were born in the US. The sample was ethnically diverse, in that just over half were non-white. Participants self-identified as Black/African American (4), South Asian (2), Latin American (2), Arab (1) white (5), and mixed/other (1).

Second, ten IPV service providers were recruited for the study for the purposes of ascertaining their understandings of empowerment, adversity, employment-seeking, and service provision. Service providers were recruited for their diverse knowledge, expertise, and experience in employment-related and economic empowerment programs. All were employed at various IPV organizations ranging from large mainstream agencies to smaller, community-specific organizations (e.g., LGBTQ, immigration, trafficking). Theoretical sampling also included recruiting service providers from economic empowerment programs and counseling programs within IPV organizations. Service providers were recruited from large, mainstream IPV organizations and smaller population-specific organizations (e.g., immigrant, LGBTQ). The inclusion criteria included those who: (1) were over the age of 18 and (2) have worked with survivors for at least three years in either a paid (e.g., social worker, advocate) or unpaid volunteer/intern capacity. All service provider participants identified as cisgender women, heterosexual (8) or queer or lesbian (2). The mean age was 36, ranging from 26 to 59. Service providers identified as white (3), Asian (3), Black or African American (3), or other/mixed race (1). Participants averaged 10 years of professional IPV experience (ranging from three to 30 years) and half of the participants were in senior management positions while the other half were front-line social workers, advocates, or economic empowerment specialists. Participants held professional degrees including MSW (7), BA (2), and MSW in progress (1).

## 7. Results

Focused analyses of the data reveal three themes: (1) structural characteristics shape individual experiences and perspectives of empowerment; (2) peer support as an integral component to empowerment; and (3) employment attainment as economic empowerment. Major themes and subthemes captured in the analysis are given in Table 1. Findings emphasize the interconnectedness of IPV and structural barriers from the perspective of both employment-seeking survivors and service providers, and supports an in-depth discussion of economic empowerment practice implications.

### 7.1. Structural Characteristics Shape Individual Experiences and Perspectives of Empowerment

Survivors and service providers were asked how they understand and view the term empowerment. Survivors' narratives revealed a range of perspectives that reflect their personal and contextualized experience.

### 7.1.1. Survivors

One survivor described empowerment as an umbrella word and discussed its wide range of interpretations in a personalized, individual context:

> Empowerment means so many things under the umbrella. Engagement, it's love, it's care, it's compassion, it's so many things. It's support, it's ideas to listen to, it's a shoulder to cry on, it's so many things. It's like an umbrella word. (Survivor participant 12).

Several survivors ($n = 10$) reflected on empowerment related to personal strength and ability to leave abuse. This survivor also described her individual process of empowerment

as a personal rediscovery and the development of resilience after the experience of violence and abuse. She put it this way:

> For me, that's what empowering means, to find who you are, and where you are going. Because literally, to be honest with you, and without any modesty. I know my future will be bright. (Survivor participant 02).

Many survivors (*n* = 9) described empowerment as a highly individualized experience that relied on self-reliance and personal strength. Conversely, while many survivors described the significance of empowerment, several found the term problematic. One survivor referred to empowerment as potentially an oppressive practice in IPV services and cautioned against service providers claiming to empower others. In some cases, survivors viewed empowerment as a process managed, or even to some extent fabricated, by service providers. This survivor described it as a social construction of white women:

> I feel it's like a term I also associate with white woman empowering women of color . . . I want to distance myself from any kind of savior model, I really want to put a lot of distance with that. (Survivor participant 08).

Many survivors commented on their challenges in economic empowerment programs (e.g., welfare-to-work, rapid job placement, shelter) that claimed to empower women. Many programs describe economic empowerment as the goal of their job training or job placement service but many had rigid rules and policies that often interfered with survivor empowerment. Several survivors highlighted the tension between individuals' perceptions of their own personal needs and the policies of programs aimed at conferring empowerment. One survivor reflected on her disempowering experiences in a Public Assistance welfare-to-work program, describing it as abusive and controlling.

> They kind of tug at you. If you make a dollar more than what they're expecting, then they can pull from you or say, oh, this is for you to handle now. But the more that you push for yourself is the more that you don't have to look to the right or the left. It's kind of like having another partner . . . Either you're going to really have a career and be self-sufficient or you're going to be reliant. And that in itself too, public assistance is kind of like a power control. (Survivor participant 11).

Another survivor shared her frustrations with a welfare-to-work program that kept her financially insecure: *"So you're working a full time job but you're getting paid less than the minimum wage because they're docking your... So you can't survive" (Survivor participant 01).*

Several survivors discussed how their intersecting identities and experiences of oppression resulted in disempowerment. A few trans survivors commented on feeling disempowered because of the lack of job opportunities due to employment discrimination and trans stigma. This survivor commented on the complexity and stigma related to her intersecting identity as trans, woman, immigrant and sex worker:

> They racial profile, they sexual profile, it happens so many times to, a few times I got arrested just for walking on the street, and they claimed that I was sex working. And I was with a friend, but my friend was white. (Survivor participant 13).

Many survivors commented on being disempowered by pervasive sexism embedded in society. As this survivor put it: *"the system is incredibly sexist and racist" (Survivor participant 01).*

Another survivor discussed the intersectional link between precarious employment, capitalism, racism, and oppression towards women's diverse social locations and experiences:

> It's more about capitalism . . . Oh, like at risk employment, we're an at risk employment state. So unless you have a recording of someone being like I'm firing you because you're Black, like you have no standing which is like also, you know, violence against, or discrimination against trans women I think. is what I hear about a lot... (Survivor participant 08).

7.1.2. Service Providers

Service providers also described their views on empowerment related to service provision with employment-seeking survivors of IPV. Their narratives also illustrate the range of perspectives on the meaning of empowerment and how perspectives can shift depending on survivors' particular experiences. Similar to some survivors, this service provider described how empowerment exists on a spectrum:

> So, empowerment is a spectrum and it depends on what is the issue, where the client is going, what the client has experienced. But it is a spectrum. (Service provider participant 09).

Service providers (*n* = 8) also viewed empowerment as a dynamic and individual process and discussed how they understood empowerment from a service provision lens. Service providers frequently used conceptual terms related to individual empowerment such as self-sufficiency, self-worth, and self-determination and discussed how they related to being empowered. This service provider describes it this way:

> I think, you know, just the whole piece of understanding your worth, your own self-worth. (Service provider participant 02).

Similar to survivors, service providers discussed personal resilience and coping. This service provider described empowerment and the role of resilience and spirituality for survivors who have overcome violence:

> I can tell you, I have never been able to see a human being as resilient as survivors of gender violence. (Service provider participant 09).

While many service providers described the importance of empowerment, some service providers, like survivors, discussed how the term is overused, disempowering, and interferes with survivors' personal agency. One service provider cautioned against the "fix it attitude" that service providers can conflate with empowerment. This service provider stated:

> I think there is a risk of so many new clinicians, especially who have this 'I can fix it' attitude, like 'I'm the expert' and I'm imparting knowledge on you. Whereas, that's not really what we're after. (Service provider participant 10).

Some service providers also commented on the problematic aspects of rapid employment placement programs that place survivors in low paying jobs. These experiences tended to be disempowering and this provider commented on negative financial consequences of these programs:

> Other workforce development programs are focused more on what we call rapid placement where it's like you do the training, you get the job and there's really not any concern about the [low] salary you're going to be paid in that job. (Service provider participant 07).

This provider further described the oppressive nature of institutions that keep survivors in a cycle of poverty

> Like a lot of systems that are designed to keep people in poverty are filled with rules and systems that make no sense, that are, you know, constantly controlling people's lives and who wants to do that. (Service provider participant 02).

Both survivors and service providers discussed the ways survivors with multiple identities (e.g., race, sexuality, gender identity, immigration status) experienced powerlessness and oppression which contributed to feeling disempowered, in the context of employment-seeking. Similar to survivors' narratives, service providers also raised concerns of sexual stigma, trans stigma, xenophobia, racism, and sexism while employment-seeking. This provider commented on trans survivors' challenge with finding employment and managing hiring bias.

> I worked with a trans person who, had, was able to get a job. And I said, "How did you do that?" And she said, "Well, I dressed as my, you know, originally assigned gender for the interview and they didn't realize so I got the job, you know. I would go in to my first day of work dressed as myself. (Service provider participant 04).

This service provider referenced the complexity of empowerment for survivors with intersecting identities and experiences of structural oppression.

> If you don't have a valid work visa it's really really hard to, you know, make the kind of wages that allow you to thrive, so you know, an immigration policy, you know, and like social policy, and racism and sexism, all those things impact the ability for you to make those [inaudible] so, um yes, I think structurally, at least in this country, you know, things have been set up to disempower, you know, marginalized groups, people of color, women . . . (Service provider participant 08).

### 7.2. Peer Support as an Integral Component to Empowerment

Nearly all survivors and service providers discussed the importance of peer support among employment-seeking survivors. Survivors described feeling empowered by sharing their IPV and employment-seeking experiences with others who could relate to their situation.

#### 7.2.1. Survivors

One survivor described feeling connected and supported by other women also seeking employment which helped her feel empowered:

> You get mentors, you get a support system, you get empowerment, you get women that is just like you and we empower each other, so you get that empowerment. You gain friendships, you gain business partners, you gain everything that your partner took away from you. That intimate partner that went through a domestic violence situation with, everything that they took from you, you gain. (Survivor participant 12).

Another survivor described feeling empowered by connecting with other women in a professional capacity who shared a similar racial background:

> Being around women who are also empowered, I feel like that shifts my whole paradigm of thinking as well. Because women who are empowered, the director for [name of department] is an African American woman at [private firm]. And that changed my whole world, just seeing someone of my color so established. (Survivor participant 11).

One survivor reflected feeling empowered through sharing her story of IPV to survivors and the non-survivors community. She described the process of sharing her personal experiences at a community-wide event:

> The fact that I had members of the community asking me questions on how to deal with domestic violence or more about my story and how I overcame it, and some other individuals like myself that made it alive out of the situation that they was in. It was very empowering, that felt really good. (Survivor participant 12).

Another survivor commented on connecting with her peers and finding community where she could belong. She reflected:

> ...creating a safe space, um like near your job or like whatever umm... I think like feeling like alone and shame and everything is just like such a big thing that like I just don't know anything that's like more helpful than community, you know? (Survivor participant 08).

Some survivors met with professionals from various job sectors who spoke about their career paths. These experiences provided an opportunity to learn about various careers and hear employment stories from female professionals. One survivor reflected on the significance of meeting women in high positions at a private banking firm.

> I would say they took us on a field trip with [Name of private firm], so that was something that really struck me, because they had like a panel of five girls and they would all say what they do, how long they've been there, their situation. So, that was very motivating to me to hear that, and also coming from successful women. (Survivor participant 05).

7.2.2. Service Providers

Service providers also observed a sense of community-based empowerment among survivors who were seeking employment. This service provider described empowerment as spanning across individual and community dimensions:

> Some individuals find their own sense of self through the work that they do as individuals initially, but some people find it by working in community and then reconnect to themselves as individuals. I've seen it kind of work both ways through the years. Either way, there's a recognition of the strength of the individual and the strength of the community. (Service provider participant 02).

One service provider also discussed empowerment related to engagement in community programs and highlighted the importance of service for women with different social locations and experiences:

> We have groups for immigrants, immigration leadership and empowerment group, we have a queer immigrant mentor's program that we just recently started to match recently arrived immigrants with immigrants who are more established in [city name], pair them up with . . . like a buddy system to navigate all of the craziness that is [city name] and have someone there to support them. (Service provider participant 01).

Some service providers described how employment programs should be conceptualized, and commented on the importance of survivor-led empowerment services. This provider commented on prioritizing survivors' role in the empowerment process:

> Making us even more survivor centered and survivor driven and then also help market that elsewhere if it's something they're interested in. But we have a lot of [inaudible] people saying, you know, "I want to be able to think about what I've gone through and how it's led me to here. (Service provider participant 04).

Another provider highlighted the importance of self-determination, autonomy, and survivor-led services:

> We really try to support people in individual work to be the ones who have their own plan, not influencing what they want but them telling us what they want to do, what goals they want to accomplish, setting out their own career plan and then we're just there to support them but they're the drivers of that plan and of what they want to accomplish in the future. So, the ability to determine their own path. (Service provider participant 01).

One service provider described the importance of peer support within economic employment programs which helped initiate survivors' personal growth and the building of a sense of community:

> A lot of people in our groups end up really supporting each other, there is an understanding around needing support and like, there is a lot of like beautiful support that happens out of some of these struggles. (Service provider participant 05).

This service provider also highlighted the importance of survivor-led services and reinforcing survivors' ability to make choices for themselves:

> So it's very survivor-led, it's like very up to the individual what they are hoping to do, so it has been a real variety of things, for one woman who was a seamstress her whole life and had run successful businesses in other countries, we made the initial investment of getting her the machinery she needed and the fabric she needed to be able to begin her business here. (Service provider participant 08).

### 7.3. Employment Attainment as Economic Empowerment

Nearly all survivor participants were engaged in a formal economic empowerment program (*n* = 11) with an IPV or community-based organization that assisted them with employment needs. Survivors reflected on economic empowerment, program experiences, and finding employment.

#### 7.3.1. Survivors

Several survivors who found employment with assistance from either economic empowerment programs within IPV organizations or outside welfare programs described their experience as empowering. As one survivor put it: *"That's how I can define empowerment, how the program empowered me" (Survivor participant 02).* Another survivor described how the hope and motivation derived from the support she received from an employment program at an IPV organization had been empowering. Becoming employed also gave her more economic security and she no longer needed welfare benefits. This survivor described her experience as follows:

> It's like for me the way that [economic empowerment program] helped me, it empowered me because it gave me hope. It motivated me, so it changed my life. It turned it 360 all the way. I have hope. I can go places now. I know that I could do better. I don't have to be stuck in PA all the time. I don't have to be in an abusive relationship. So, it did empower me because it gave me hope. (Survivor participant 05).

Another survivor described how employment services offered by welfare programs and IPV organizations helped them find employment. She described achieving her goal to go back to school and reflected:

> Once you study, and you see that you're understanding something, and you're passing the tests, and then you're finished the college and you get a good job, this all is like, gives you . . . adds to your confidence about yourself. (Survivor participant 14).

Some survivors also discussed interpersonal/relational dimensions of empowerment through a process of gaining knowledge, education, or skills from an educator or service provider within an economic empowerment program. This survivor discussed this significance:

> In my case, empowerment was knowledge . . . I think education and knowledge is the key to be empowered . . . Whether you're straight, gay, trans. (Survivor participant 13).

Many survivors discussed economic empowerment as it relates to gaining employment. One survivor reflected: *"So, that's the goal for women to be empowered and have a salary" (Survivor participant 11).* She engaged in an economic empowerment program and got a job and reflected on that experience of gaining employment and empowerment:

> Employment is empowering for itself because when a woman is able to do for themselves or do for their loved ones, it gives her a sense of self. It makes her know that she is important, she has identity, and that she has something to contribute to the world. And when you're able to know within yourself that there's a purpose, you have a reason to live with purpose. (Survivor participant 11).

Not all survivors gained employment after completing an employment program. Some survivors were job-seeking and others were engaging in an internship or higher education. One woman described an employment program that did not help her find employment: *"They didn't find me a job, but it helped with my self-esteem"* (Survivor participant 10).

### 7.3.2. Service Providers

Several service providers (*n* = 10) provided formal and informal employment services and supported survivors within IPV organizations. Service providers described their perspectives and discussed the link between empowerment, employment, and the benefit of employment services. This provider stressed the importance of personal agency in economic empowerment services:

> So helping somebody like that, figure out a) what they want to do, find their own agency, do that [inaudible] and also then, you know, navigate all the things that come with that, you know, from negotiating salaries to kind of the interpersonal relationships on the jobs to, you know, all of the different kind of things that go into work, that we take for granted if we haven't been exploited, has been a real focus for a lot of my work. (Service provider participant 08).

Service providers discussed economic empowerment related to goal attainment such as achieving financial security and seeking employment. Some service providers linked becoming empowered to having financial security. One provider described it as the ability to be in control over one's financial well-being.

> Obviously, intimate partner violence can also be financial violence and getting people support, as we were talking about, feel empowered to be financially stable independent of a partner is also really important, getting these pieces in place to have support. (Service provider participant 01).

This provider commented on the interpersonal/relational process of economic empowerment as an interaction generated between provider and survivor and described it this way:

> We are focused on economic empowerment . . . And are really helping clients kind of craft plans that will allow them to build the necessary skills that will allow them to gain the necessary knowledge to begin or resume career that have been stymied by their experience of domestic violence or gender based violence. (Service provider participant 07).

Some service providers described working towards empowerment and social change by helping educate and raise awareness about employment discrimination among employers. One service provider highlighted the importance of addressing employment discrimination at a structural level for trans survivors:

> We train workforce providers and employers to how they can create a more affirming environment for trans staff members once they hire trans staff members. And then on a more basic level, what does it mean to be trans, what's the language around that, what are some best practices for hiring and for interviews and treating everyone with respect? Basic stuff that gets people used to language and learning about how to talk about LGBTQ issues. (Service provider participant 01).

This service provider also linked economic empowerment efforts at the structural level and described her program:

> The [organization] just started a policy and advocacy program and we're building on that and getting community feedback and channeling community members into these different areas where they can really try and make some change so not having people just feel like, okay, I am a part of this system, but actually being able to feel like they're empowered to make a difference and make change. (Service provider participant 01).

Another service provider also described her efforts this way:

Larger social structures really need to be challenged and policy needs to explicitly protect the, you know, the um right of all people. And I think when we talk about racism and we talk about, you know, immigration, and we talk about all of these things, there's no way to separate them from empowerment in this country. (Service provider participant 08).

This service provider considered the different and unique experiences of empowerment and considered strategies to support clients. She described it as a highly individualized process that aims to help survivors heal:

You know, there are different ways of finding one's empowerment, because we are different so there should be different methods, different strategies, different tools. And I think helping our clients see that there's no one way of healing themselves, that there are many different ways. There are many different ways that they have experienced the abuse so there has to be many different ways of healing from it. (Service provider participant 02).

## 8. Discussion

Survivors gave their unique and personal perspectives as recipients of empowerment and employment services, while service providers discussed their views broadly based on their professional knowledge and experiences. Both groups described the positive associations with empowerment related to employment-seeking after leaving abuse (e.g., living violence-free, connecting with other survivors, gaining employment) as well as negative associations with empowerment (e.g., oppressive practices, economic insecurity, structural barriers that impede empowerment). Generally, views on empowerment and employment-seeking aligned among service providers and survivors. However, there were some inter-group differences which at times resulted in opposing viewpoints among survivors and service providers. This produced several pointed observations as to what constitutes empowerment, who is responsible for empowering survivors, and how economic empowerment programs should be conceptualized for employment-seekers.

Survivors of IPV conceptualized empowerment as a highly individualized process of transformation and recounted personal experiences of resilience and personal growth, especially after leaving abusive relationships. This finding is in line with previous research investigating IPV and career counseling interventions [14]. Service providers also discussed the individual aspects of empowerment, but their insights focused on particular service strategies and approaches to help facilitate economic empowerment. Service providers frequently referenced conceptual terms such as self-sufficiency, self-efficacy, self-determination, and individual agency as empowerment-related outcomes.

However, some service providers questioned the focus on individual outcomes that rely on traditional views of neoliberalism rooted in an individual conceptualization of empowerment [23,24]. Findings in this study reveal that employment-related services may be overly individualistic, a characteristic that obfuscates the structural determinants of empowerment. For example, both groups described certain social welfare programs and rapid job placement practices as overly focused on individual outcomes (e.g., attendance in low skills training program, securing any low paying job). This was described by both survivors and providers who were concerned that certain programs had the potential to further disempower survivors due to the focus on individual success in the absence of structural factors. Rigid program policies, victim-blaming attitudes, and the inability to support survivors in finding living wage jobs are some examples of disempowering experiences reported by survivors. With an increasingly unstable labor market and experiences of marginalization, IPV survivors are incrementally vulnerable to experiencing insecure or precarious employment and poverty [40].

Furthermore, there were some important intra-group differences with respect to viewpoints regarding personal agency and autonomous decision-making tied to empowerment-

related services. While some survivors and service providers highlighted the importance of personal agency, others commented on a lack of agency that further oppressed survivors seeking employment. In some cases, survivors were unable to exercise agency because others (e.g., service providers, welfare policies) actually held the power with respect to ensuring that survivors received specific job placement or training programs. Both groups recommended that service providers should avoid reproducing dominant-subordinate power dynamics in their relationships with survivors such as assuming the role of experts or assuming responsibility to empower clients, both of which interfered with personal agency. One survivor highlighted the racial overtones of empowerment and associated it with "*white women empowering women of color*," relating it back to the earlier discussions of empowerment as a construction of white middle-class women [41]. This finding highlights tensions between program efforts and institutional policy that aim to confer empowerment.

However, some survivors appreciated the expertise of knowledgeable service providers, especially when services were seen as inclusive, client-centered, and focused on attaining living-wage employment. Survivors' experiences in economic empowerment groups at IPV organizations were also viewed as beneficial and helped initiate both personal and collective growth while providing an opportunity to learn employment skills. Both survivors and service providers offered recommendations for empowerment-focused employment programs. While focusing on individual empowerment outcomes is essential in some instances (e.g., enhancing self-esteem), it is not enough for a fulsome approach to economic employment practices. Many survivors and service providers emphasized the importance of community and peer support which was empowering for survivors seeking employment. Peer support was also key in that it enhanced a sense of membership in a community of survivors which, in turn, helped to alleviate a sense of isolation, a finding that aligns with previous research [42,43]. Several survivors commented on the importance of growing their professional networks through their relationships with survivor peers. Many survivors had been cut off from professional networks as a result of the abusive relationship and connecting with other survivors in a professional context was perceived as empowering. This finding has been reported in prior studies on IPV and group career counseling [14,20].

Survivors and service providers both commented on the diverse and intersecting identities and experiences of marginalization that impacted employment-seeking. Intersectionality and stigma (sexual and trans stigma) influenced their experiences, creating a disempowerment that interfered in their employment. Survivors with particular identities (trans, immigrant, women of color) and survivors with intersecting combinations of these identities had very complex experiences of oppression (e.g., racism, sexism, hiring bias, employment discrimination) which overlap, shape and exacerbate each other. Survivors identified many of the discriminatory hiring practices adopted by employers, and described how they created significant structural obstacles to their economic empowerment at the structural level. In contrast, some service providers gave examples of structural empowerment through advocacy endeavors that challenged inequities and oppression. Service providers identified a range of social change efforts (e.g., working with human rights groups, connecting with other IPV organizations, working with employers). This finding can be considered another important contribution of the present study in that it highlights the critical significance of social action strategies as a form of economic empowerment which challenges oppressive practices and policies due to gendered violence, class, race, sexual identity, and immigration status. This is also a significant element of the original formation of IPV services which valued social change efforts, one which can be applied in the context of supporting survivors who are employment-seeking.

Study findings provide a platform for recommendations on the next steps to establish new approaches to economic empowerment service provision. For example, effective economic empowerment services should focus on individual employment needs of survivors, include more autonomous/survivor-led opportunities, involve elements of peer support, emphasize social change, and raise awareness of structural inequities [44]. This finding is important because it demonstrates the range of perspectives on empowerment that consid-

ered individual/intrapersonal processes, relational aspects, community empowerment, and structural oppression. It also shows how economic empowerment services are responding to employment needs of survivors while considering social, political, and cultural contexts.

The perspectives of survivors and service providers on empowerment and employment-seeking can be understood through various theoretical lenses. For example, some survivors felt they needed more individual level empowerment support, while other service providers prioritized structural issues that interfered with employment-seeking. This paper builds on current IPV and employment research by proposing a perspective that incorporates a social ecological approach within an intersectionality framework. This approach challenges current sociopolitical limitations of theories of empowerment, as it broadens conceptualizations to emphasize the interconnectedness of personal, interpersonal, and political domains of empowerment and employment-seeking. In doing so, empowerment-oriented practices for employment-seeking take into account each of the individual (intrapersonal/psychological), interpersonal (relational), community, and structural (macro) components of the empowerment process [16,25,32,43,45]. Table 2 provides a detailed outline of each level of empowerment and related practices. Further research that examines the interconnectedness of experiences of IPV, employment-seeking, and structural determinants related to survivors' identities and experiences of IPV and empowerment is needed to better support employment-seeking that can lead to successful work outcomes [46].

**Table 2.** Economic empowerment practice examples for employment-seeking survivors of intimate partner violence (IPV).

| Type of Economic Empowerment | Definition | Examples |
|---|---|---|
| Individual empowerment | Individual, intrapersonal or psychological empowerment refers to the individual level of empowerment and personal processes of growth and change [32]. | • Increasing individual agency, confidence, self-esteem<br>• Raising awareness on experiences of violence, personal resilience, increasing decision-making |
| Relational empowerment | Relational or interpersonal empowerment emphasizes the importance of relationships and focuses on power dynamics, domination, liberation, and reciprocity. This type of empowerment values the importance of establishing relationships [47]. | • Individual career counseling, job search assistance (e.g., resume, interview preparation),<br>• Peer support, mentorship, opportunities for skills-building<br>• Considerations of power dynamics in relationship, reciprocal empowerment processes. |
| Community empowerment | Community empowerment involves a group of individuals who share a common experience or identity. Community empowerment generally occurs in a group setting and provides opportunities for personal growth and social change [43]. | • Group career counseling, group training seminars, professional development events, skills building workshops, public speaking, consciousness-raising activities, community mobilization, personal growth and community building concurrently, social change. |
| Structural empowerment | Structural empowerment seeks to address structural inequities experienced collectively by groups who have experienced oppression. This can also include systemic issues such as economic insecurity, lack of services, policies and practices that impact communities that experience oppression [48]. | • Advocacy-related activities, engage in social action efforts, increase access to employment programs, challenge employment discrimination, address oppressive structures, train employers on issues that impact survivors (trauma, violence, discrimination). |

## 9. Implications for Policy and Practice

The study has several implications for policy which should be understood within the current sociopolitical climate; it illuminates the implications of larger and intercon-

nected structural forces such as the impact of neoliberalism on social welfare, and labor laws and policies, and the political climate in the U.S. One of the major repercussions of neoliberalism on IPV services is reduced government spending on social services and increased responsibility of the private sector to find solutions to fund services [23]. Many IPV organizations experience funding cuts to programs and struggle to find resources to support existing services [49]. In many cases, IPV organizations are forced to scale back on programs and reduce amount of sessions that survivors can attend [25]. IPV organizations and service providers are responding to and looking for innovative ways to respond to service needs [23,42]. This is evident from the study findings. Service providers are addressing the gap in employment services by finding creative ways to learn about career counseling, small business training, and other employment services, often with little training or educational background. However, there is a need for more leadership, funding, and partnerships to support these programs.

Neoliberal ideology continues to be embedded in services that focus on micro level change that centers on individual accountability. This perspective frequently ignores structural level factors or the impact of structural oppression on clients [23]. This ideology assumes a survivor should get a job immediately after leaving an abusive relationship and should work hard to support herself and children without acknowledging, recognizing, or redressing structural barriers. This assumption shows a blatant disregard for women who have complex experiences of violence, and who are recovering from the emotional, physical, economic effects of IPV; it does not recognize the diversity of backgrounds and identities of survivors of IPV [25]. Furthermore, this perspective does not take into consideration the complex experiences of oppression and structural level violence (e.g., racism, classism, transgender stigma, sexual stigma) [23] which can influence a survivor's ability to find a job. This is evident in the narratives of survivors and service providers in this study who held multiple identities, backgrounds, and unique experiences. Most of the participants were women of color, more than half were immigrants, and some identified as queer or trans. Their experiences of violence also varied (e.g., interpersonal violence, trafficking) and revealed complex employment-seeking experiences. Individualized services should focus on improving survivors' employability, recognizing their social locations, addressing multilevel barriers to employment—and prioritize their specific needs, goals, and preferences. Findings support that employment services do not fit the one-size-fits all model and survivors require individualized services tailored to their unique needs rather than traditional neoliberal models.

Securing work immediately after leaving an abusive relationship is a challenge for many survivors. Many survivors are living in crisis shelters, caring for children and recovering from the physical and emotional impact of abuse [50]. These conditions make securing work after leaving an abusive relationship a very difficult proposition for some survivors. Additionally, rapid job placement does not consider the importance of living wage employment and the ability of the survivor to secure a job that provides stability, security, and growth [25]. Findings from the present study have implications for labor and welfare policies, specifically disputing welfare programs that require recipients to find immediate employment. Policies should strive to facilitate access to employment initiatives that focus on long-term career development and living wage employment, taking into consideration the broad impact of IPV on survivors' employment.

Findings from this study also illuminates multiple violations of labor laws including blatant employment discrimination against survivors and their various identities, as cited by both survivors and service providers. For example, more than half of participants had children in their primary care. Several participants reported that potential employers inquired about their children and childcare arrangements and that they were overlooked for jobs once they reported their single-parent status. This obvious violation of labor laws requires further investigation in employment hiring policies and procedures. Further advocacy is also called for in this area to protect survivors who have children and are

seeking employment but who are being discriminated against because of their parent status [51].

Additionally, survivors who identified as queer or trans faced significant discrimination and prejudice while employment-seeking. One out of four people in the LGBTQ community still experienced job discrimination based on their sexual orientation or gender identity and nearly a third of trans people were fired or not hired as a result of their trans status [52]. Title VII of the Civil Rights Act of 1964 prohibits any employment discrimination based on race, religion, national origin and sex. A case which would establish whether LGBTQ individuals are covered under the Civil Rights Act was before the U.S. Supreme Court in 2019 [53]. Some states and the Equal Employment Commission (EEOC) have applied this law to LGBTQ individuals but there remain several states that offer no protection against employment bias and discrimination [54]. Additionally, despite some state protection, findings in this dissertation highlight that LGBTQ survivors continue to face job discrimination as a result of their sexual orientation and gender identity even in states that have laws in place. This reality is alarming and requires immediate action and further anti-discrimination laws in order to protect LGBTQ communities from victimization. More public attention and advocacy efforts are required to protect LGBTQ survivors from being fired or harassed at work and overlooked for employment [55]. Service providers and advocates need to ensure nondiscrimination and inclusion in the workplace through promoting LGBTQ-inclusive workplace policies, identifying employers who are inclusive/supporting employers to become more inclusive, and funding of LGBTQ employment support programs.

## 10. Limitations

This is one of the first studies to explore economic empowerment from the perspectives of IPV survivors and service providers in the context of employment-seeking. However, there are some limitations to consider within the context of the study methods. In-depth interviews provide important insights but there are limitations related to the sample size, study location and demographics and these preclude transferability of the findings to all IPV survivors. Survivor participants were recruited from IPV organizations where they were actively seeking employment and counseling services in a large, urban city. Survivors who were not actively employment-seeking were not considered for this study; as such, it was not possible to compare findings with those who did not engage in formal employment services. Future research would benefit from a greater number of participants, more representation of survivors with different abilities, and more geographic diversity. Additionally, a comparative study of survivors engaged in employment programs and those who are not, a study that also collects data on long-term job retention, salary, and career changes would also be beneficial. Despite these limitations, the strengths of the study include a qualitative approach that explored both the experiences of IPV survivors and service providers and multiple dimensions of empowerment and employment-seeking.

## 11. Conclusions

The objective of this study was to compare perspectives on economic empowerment among employment-seeking survivors of IPV survivors and service providers. The unique views shared by survivors and service providers on IPV, employment-seeking, and empowerment contributed to increasing the limited knowledge in the field. Views frequently aligned between groups, although differences emerged within both the survivor and service provider group. Some service providers and survivors subscribed to more of a structuralist point of view consistent with a multidimensional understanding of empowerment, while others considered economic empowerment as an attribute of the individual. This tension provided insights into what constitutes empowerment, who is responsible for empowering survivors, and how economic empowerment programs should be conceptualized for employment-seekers. Institutions and programs which are limited by individual-level notions of empowerment based on neoliberal models can contribute to

the systemic oppression of employment-seeking survivors. By expanding individual level empowerment models and including an intersectional understanding of multiple structural barriers (e.g., gender, race, socio-economic, sexuality, and immigration), economic empowerment services may better meet the employment needs of survivors. Survivors seeking living wage employment—the key to economic freedom and economic empowerment—are best supported by economic empowerment programs that utilize an inclusive and client-centered approach, provide opportunities to connect with peers, and address structural level determinants.

This study provides insight into the individual perspectives on empowerment and employment-seeking of survivors of IPV and service providers. Their views should provoke discussions about individual empowerment, structural determinants of social position, living wage employment in the labor market; they also provide important practice considerations. First, this study highlights the importance of expanding the manner in which service providers conceptualize economic empowerment. Current economic empowerment programs are frequently individualistic and have the potential to disempower women due to the focus on individual success at the expense of structural factors. Practitioners can build on the strengths of individually-focused economic empowerment (e.g., employment attainment, increase in self-esteem/confidence) and integrate a more structural perspective on empowerment to better achieve the goal of empowering women through financial independence by way of employment. Integrating a structural perspective includes addressing the discrimination faced by women of color, trans women, and immigrants during the employment process through advocacy with potential employers. It also includes communicating with public policy officials and those responsible for designing service programs to urge them to be more inclusive of survivors and their diverse and intersecting identities.

This study also illustrates the multi-layered vulnerability associated with intersecting axes of social position, an area of research that is highly relevant to understanding the challenges associated with achieving economic empowerment in vulnerable populations [56].

Findings signal that empowerment continues to be central to IPV services and important to survivors as they heal from abuse. However, service providers need to remain cautious and ensure that services are not oppressive or victim-blaming. Providers can adopt a more comprehensive understanding of economic empowerment that considers the intersectional identities of survivors, identifies their multilevel service needs (e.g., individual autonomy, survivor-led services, peer support, structural determinants, social change) and ensures they are engaged and have agency over the manner in which they receive services.

**Author Contributions:** Conceptualization, S.T., H.S.-M. and R.A.; methodology, S.T. and R.A.; formal analysis S.T.; writing—original draft preparation, S.T.; writing—review and editing, S.T., H.S.-M. and R.A.; supervision, R.A. and H.S.-M. All authors have read and agreed to the published version of the manuscript.

**Funding:** This research received no external funding.

**Institutional Review Board Statement:** Institutional Review Board Statement and approval number #35337.

**Informed Consent Statement:** Informed consent was obtained from all subjects involved in the study.

**Data Availability Statement:** The data are not publicly available due to the privacy and ethical issue.

**Conflicts of Interest:** The authors declare no conflict of interest.

## Appendix A

**Table A1.** IPV survivor demographics (*n* = 16).

| Demographic Information | Participants (%) |
|:---:|:---:|
| **Age** | |
| 20–29 | 5 (31) |
| 30–39 | 6 (38) |
| 40–49 | 4 (25) |
| 50–59 | 1 (6) |
| **Gender** | |
| Cisgender female | 13 (81) |
| Trans female | 3 (19) |
| **Sexual Orientation** | |
| Straight | 12 (75) |
| LGBTQ | 4 (25) |
| **Ethnicity/Cultural Background** | |
| White/European | 5 (31) |
| Black/African American | 4 (25) |
| Hispanic/Latin American | 2 (13) |
| South Asian | 2 (13) |
| East Asian | 1 (6) |
| Arab | 1 (6) |
| other | 1 (6) |
| **Country of Birth** | |
| USA | 5 (31) |
| Other | 11 (69) |
| **Highest Completed Educational Level** | |
| High school/GED | 6 (38) |
| Some college | 2 (13) |
| Bachelor | 6 (38) |
| Master | 2 (13) |
| **Number of Children** | |
| 0 | 6 (38) |
| 1 | 5 (31) |
| 2 | 0 |
| 3 | 1 (6) |
| 4 | 0 |
| 5 | 1 (6) |
| **Household Income Range** | |
| Under $10,000 | 4 (25) |
| $10,000–$19,999 | 4 (25) |
| $20,000–$29,999 | 3 (19) |
| $30,000–$39,999 | 1 (6) |
| $40,000–$49,000 | 3 (19) |
| **Employment Status** | |
| Employed | 10 (63) |
| Part-time | 6 (38) |
| Full-time | 4 (25) |
| Unemployed | 6 (38) |
| **Occupation** | |
| Beauty | 1 (6) |
| Non-profit/education | 3 (19) |
| Food service | 1 (6) |
| Private (legal, financial firms) | 2 (13) |
| Health | 2 (13) |

Table A2. IPV service provider demographics.

| Demographic Information | Participants (%) |
|:---:|:---:|
| **Age** | |
| 20–29 | 2 (20%) |
| 30–39 | 5 (50%) |
| 40–49 | 2 (20%) |
| 50–59 | 1 (10%) |
| **Gender** | |
| Female (cisgender) | 10 (100%) |
| **Race** | |
| White | 3 (30%) |
| Asian | 3 (30%) |
| Black/African American | 3 (30%) |
| Mixed | 1 (30%) |
| **Sexual Orientation** | |
| Heterosexual | 8 (80%) |
| Queer/Lesbian | 2 (20%) |
| **Country of Birth** | |
| USA | 8 (80%) |
| other | 2 (20%) |
| **Highest Completed Education** | |
| MSW | 7 (70%) |
| MSW in progress | |
| BA | 2 (20%) |
| **Years Working in IPV** | |
| 0–3 | 1 (10%) |
| 4–7 | 4 (40%) |
| 7-10 | 2 (20%) |
| 10 and up | 3 (30%) |
| **Types of IPV Organization** | |
| Large, mainstream | 5 (50%) |
| LGBTQ | 2 (20%) |
| Trafficking, system impacted | 2 (20%) |
| Financial/Economic Empowerment | 1 (10%) |
| **Current Position** | |
| Director | 5 (50%) |
| Supervisor, manager or coordinator | 3 (30%) |
| Case manager/advocate | 2 (20%) |

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
