# Peer review of "An Analysis of Comparative Perspectives on Economic Empowerment among Employment-Seeking Survivors of Intimate Partner Violence (IPV) and Service Providers"

_societies, doi:10.3390/soc12010016_

Round 1

Reviewer 1 Report

Kindly adhere strictly to the comments provides for better enhancement of the work.

Author Response

Comments to Reviewer #1

We thank the reviewer for the thoughtful comments and after careful consideration and consultation with colleagues we have decided to preserve the original text for clarity.

Reviewer 2 Report

This research deals with intimate partner violence (IPV). The novelty of this research is based on providing evidence on economic empowerment in the context of employment seeking among survives and service providers specializing in IPV trauma.

The manuscript is well justified and organized, with up-to-date cites. The analyses conducted are appropriate and well described.

Some issues are described in the case of authors would consider them:

In the abstract section, I recommend that the research question be included in the first paragraph (purpose) and the authors will describe instruments in the method paragraph. Also, I recommend that analyses will be described in the results paragraph.

In the introduction section, the authors should describe the prevalence of IPV.

Secondly, I consider that the aim of the study (lines 67- 72) will be included in the last paragraph of this section because the authors had explained the background of the topic and after they concluded the objective of this research.

I find a style mistake in line 120, there is a dot and space before cite. Moreover, this sentence  “As outlined in Chapter 2, these requirements pose several problems” in line 134 is confusing. Are the authors referring to the chapter of PRWORA?

In the method section, the sample description could be included in a Table.

I consider that the sample description section will be indicated before the data analysis section. On the other hand, the inclusion criteria should be revised. The authors include information about the inclusion criteria and characteristics of the sample (lines 258-267). I understand that lines 215-217 refer to inclusion criteria, but the authors only indicated some of them. And, the sample characteristic had been explained in the sample description section.

In the data analysis section, the authors should describe the statistical analysis in more detail.

In the results section, I recommend that authors will include more information about de sample, for example: “Several service providers (n=10)”.

In the conclusion section, the authors could indicate the aim of their study in the first paragraph and the following paragraph explains the main results.

Author Response

This research deals with intimate partner violence (IPV). The novelty of this research is based on providing evidence on economic empowerment in the context of employment seeking among survives and service providers specializing in IPV trauma.

Thank you for your feedback, we really appreciate it

The manuscript is well justified and organized, with up-to-date cites. The analyses conducted are appropriate and well described.

Thank you

Some issues are described in the case of authors would consider them:

In the abstract section, I recommend that the research question be included in the first paragraph (purpose) and the authors will describe instruments in the method paragraph. Also, I recommend that analyses will be described in the results paragraph.

Thank you. The authors have included this information in the abstract

In the introduction section, the authors should describe the prevalence of IPV.

Thank you. Given we indicate that IPV impacts 1 in 3 women worldwide, we feel this should be sufficient

Secondly, I consider that the aim of the study (lines 67- 72) will be included in the last paragraph of this section because the authors had explained the background of the topic and after they concluded the objective of this research.

Thank you, we have made this correction.

I find a style mistake in line 120, there is a dot and space before cite. Moreover, this sentence  “As outlined in Chapter 2, these requirements pose several problems” in line 134 is confusing. Are the authors referring to the chapter of PRWORA?

Thank you, we have fixed this.

In the method section, the sample description could be included in a Table.

Thank you, we have included a sample description of both participant groups in a table

I consider that the sample description section will be indicated before the data analysis section. On the other hand, the inclusion criteria should be revised. The authors include information about the inclusion criteria and characteristics of the sample (lines 258-267). I understand that lines 215-217 refer to inclusion criteria, but the authors only indicated some of them. And, the sample characteristic had been explained in the sample description section.

Thank you, we have included a statement on inclusion criteria for survivor participants

In the data analysis section, the authors should describe the statistical analysis in more detail.

Thank you. This is a qualitative study and no statistical analysis were used.

In the results section, I recommend that authors will include more information about de sample, for example: “Several service providers (n=10)”.

Thank you. We have added a bit more information about the sample size in some sections.

In the conclusion section, the authors could indicate the aim of their study in the first paragraph and the following paragraph explains the main results.

Thank you, the aim of the study has been included in the first paragraph of the conclusion.

Round 2

Reviewer 1 Report

It is okay now

Reviewer 2 Report

Dear Editor and Authors,

I considered that the manuscript is improved. Regrets to note that the authors have indicated that they included in the new version of the manuscript a table with a sample description of the participants. However, I do not find this table. I recommend you review this manuscript.